

# Measurement Report: Hygroscopicity and mixing state of submicron aerosols in the lower free troposphere over central China: local, regional and long-range transport influences

Jingnan Shi[1,★], Zhisheng Zhang[2,★], Li Li[3], Li Liu[4,#], Yaqing Zhou[5,6], Shuang Han[7], Shaobin Zhang[5,6], Minghua Liang[5,6], Linhong Xie[5,6], Weikang Ran[3], Shaowen Zhu[5,6], Hanbing Xu[8], Jiangchuan Tao[5,6], Alfred Wiedensohler[9], Qiaoqiao Wang[5,6], Qiyuan Wang[3], Nan Ma[5,6], Juan Hong[5,6,#]

[1]Institute of Facility Agriculture of Guangdong Academy of Agricultural Sciences, Guangzhou, 510640,
China
[2]Guangdong Provincial Key Laboratory of Water and Air Pollution Control, South China Institute of Environmental Science, Ministry of Ecology and Environment, Guangzhou, 510655, China
[3]State Key Laboratory of Loess and Quaternary Geology, Key Lab of Aerosol Chemistry and Physics, Institute of Earth Environment, Chinese Academy of Sciences, Xi'an, 710061, China
[4]Guangzhou Institute of Tropical and Marine Meteorology of China Meteorological Administration, GBA Academy of Meteorological Research, Guangzhou, 510640, China
[5]Institute for Environmental and Climate Research, Jinan University, Guangzhou, 511443, China
[6]Guangdong-Hongkong-Macau Joint Laboratory of Collaborative Innovation for Environmental Quality, Guangzhou, 511443, China
[7]Department of Geography, College of Science, Qiqihar University, Qiqihar 161006, China
[8]School of Computer Science and Engineering, Sun Yat-Sen University, Guangzhou, 510006, China
[9]Institute for Tropospheric Research, Permoserstr. 15, Leipzig, 04318, Germany
★These authors contributed equally to this work.

*Correspondence to*: Juan Hong (juanhong0108@jnu.edu.cn) and Li Liu (liul@gd121.cn)

**Abstract.**

Understanding the hygroscopicity and mixing state of atmospheric aerosol particles is crucial for improving predictions of cloud formation and climate impacts. However, measurements in the lower free troposphere - a representative atmospheric layer characterizing regional background conditions in aerosol transport and atmospheric evolution - remain sparse, especially in regions influenced by both
anthropogenic emissions and long-range transported air masses. This study adds further data on size-resolved hygroscopicity and mixing state measurements of aerosols at Mt. Hua (2060 m a.s.l., central China) during October-November 2021 using a Hygroscopicity Tandem Differential Mobility Analyzer





(HTDMA). Results demonstrate size-dependent hygroscopicity, with the mean hygroscopicity parameter ($\kappa_{mean}$) increasing from 0.20 (30 nm) to 0.30 (200 nm). The ambient submicron aerosols were primarily externally mixed, dominated by more-hygroscopic (MH) particles, with no significant diurnal variation, indicating minimal influence from boundary layer dynamics. Aerosols originating from Mongolia deserts tended to be less hygroscopic, associated with an enhanced number fraction of less-hygroscopic (LH) mode particles relative to those from other sources. However, during episodes of striking high relative humidity (RH > 80%), atmospheric aerosols containing mineral dust showed unexpected hygroscopic enhancement, suggesting in situ RH-driven chemical processing that increased aerosol hygroscopicity. Atmospheric aerosols at Mt. Hua displayed distinct hygroscopic properties compared to other high-altitude sites, underscoring regional differences in aerosol sources and free tropospheric processing. These findings advance our understanding of aerosol aging and processes in the lower free troposphere over central China, and offer crucial observational constraints for modeling aerosol–cloud interaction and regional climate impacts.

## 1 Introduction

Aerosol hygroscopicity, defined as the ability of aerosol particles to absorb water vapor from the surrounding atmosphere (Cai et al., 2017; Petters and Kreidenweis, 2007; Swietlicki et al., 2008), is a critical property that influences Earth's radiative balance by altering particle size distributions and associated optical properties (Bai et al., 2018; Cheng et al., 2008; Hong et al., 2018; Su et al., 2010). This property also plays a pivotal role in cloud processes by modulating the concentration of cloud condensation nuclei (CCNs) and affecting the lifetime and microphysical properties of clouds, thereby exerting indirect effects on regional and global climate (Liu et al., 2013; Rosenfeld et al., 2014). Additionally, aerosol hygroscopicity can regulate heterogeneous and multiphase chemistry through promoting the uptake reaction rates of different trace gases (e.g., $SO_2$, $NO_X$) within hydrated aerosols with elevated aerosol liquid water content (Tong et al., 2020; Wang et al., 2020; Wu et al., 2018b). Beyond atmospheric impacts, it also governs respiratory health by determining the deposition efficiency and site of inhaled particles within the human respiratory tract (Farkas et al., 2022).

High-altitude alpine regions in the free troposphere, distanced from local pollution yet strongly influenced by long-range transport, serve as representative of the atmospheric characteristics on a large-



scale (Colbeck et al., 1998). Additionally, these regions can also signify the interactions between the free troposphere and the planetary boundary layer (PBL) through vertical advection of lowland pollutants during the diurnal PBL evolution (Holmgren et al., 2014). Despite these significance, comprehensive measurements of aerosol hygroscopicity in such high-altitude environments still remain scarce, with only

very few studies available to date. For instance, observations at Jungfraujoch, Switzerland (3580 m a.s.l.), revealed that injections of PBL air reduced the hygroscopicity of the background free-tropospheric aerosols (Kammermann et al., 2010). Similarly, aerosols measured at the Monte Cimone Observatory in Italy (2165 m a.s.l.) showed similar hygroscopicity compared to Jungfraujoch aerosols, though occasionally influenced by the long-range-transported African dust (Dingenen et al., 2005). At the

slightly lower-altitude Puy de Dôme station (1465 m a.s.l.), aerosols displayed intermediate hygroscopicity due to mixed influences from oceanic, continental, African and local air masses (Holmgren et al., 2014).

However, these limited observations have been largely confined to European alpine sites, leaving a critical gap in understanding aerosol hygroscopicity in other high-altitude regions, particularly in Asia,

where atmospheric conditions and pollution sources differ substantially. Mt. Hua (2060 m a.s.l.), located in central China, presents an ideal setting for such investigations due to its unique geographic position. The mountain lies adjacent to the heavily polluted Guanzhong Basin, which experiences intense anthropogenic emissions, while also being downwind of the vast Mongolian deserts. This particular exposure makes the site susceptible to both regional pollution plumes and frequent dust intrusions, most

likely resulting in a more complex particle composition compared to typical ground-level measurements. It has been documented that (Wang et al., 2012) mineral dust levels in aerosols at Mt. Hua increased sharply during dust storms originated from the Mongolian deserts. Notably, this mineral dust can undergo chemical modification after transport through polluted regions, interacting with other anthropogenic pollutants. Such processes may further complicate the aerosol composition that ultimately reached Mt.

Hua and consequently alter the hygroscopic properties of particles at the site. However, the combined influence of these mixed sources on aerosol hygroscopicity at Mt. Hua has yet to be systematically evaluated.

In this study, we present the first direct measurements of aerosol hygroscopicity and mixing state at the summit (~2060 m a.s.l.) of Mt. Hua during early winter of 2021, using a self-assembled

Hygroscopicity Tandem Differential Mobility Analyzer (HTDMA). To our knowledge, this work



provides the first comprehensive investigation of aerosol hygroscopicity at a high-altitude site in central China. By analyzing the effects of diurnal cycles and air mass types on aerosol hygroscopicity, we gain new insights into the atmospheric processing and regional evolution of background aerosols.

## 2    Site and measurement methods

### 2.1 Site description

The field campaign was conducted at the summit of Mt. Hua (34°28′N, 110°05′E; 2060 m a.s.l), located in the central part of the Guanzhong Basin of western China, from October 15 to November 19, 2021. The sampling area was predominantly covered with vegetation and was distant from anthropogenic influence, with the nearest city (Huayin) being about 5.9 kilometers away from the mountain base. Being

mostly situated within the free troposphere as suggested earlier (Shen et al., 2023), this site effectively represent the free tropospheric background atmospheric conditions in this region.

### 2.2   Aerosol hygroscopicity measurements and data analysis

A self-assembled HTDMA system was used to characterize aerosol hygroscopicity during this study. A detailed description and the working principles of the HTDMA system can be found in previous studies

(Han et al., 2022; Tan et al., 2013). Specifically, the hygroscopic growth factor (GF) was measured for particles with dry diameter of 30, 60, 100, 150 and 200 nm at 90 % relative humidity (RH). The system was calibrated once a week using ammonium sulfate, which is a reference material with well-characterized hygroscopic properties.

In this study, the hygroscopicity parameter, $\kappa$, introduced by Petters and Kreidenweis as (Petters and

Kreidenweis, 2007), was used to describe the hygroscopicity properties of aerosol particles based on the measured GF as :

$$\kappa = (GF^3 - 1)\left(\frac{\exp\left(\frac{A}{D_{dry}GF}\right)}{RH} - 1\right),$$

(1)

$$A = \frac{4\sigma_{s/a}M_w}{RT\rho_w},$$    (2)

GF is the hygroscopic growth factor measured by HTDMA at 90 % RH. $D_{dry}$ is the size of the



particles. $\rho_w$ and $Mw$ are the density and molecular weight of water. $\sigma_{s/a}$ is the surface tension of the

droplets, which is assumed to be that of pure water ($\sigma_{s/a} = 0.0728$N m$^{-2}$). $R$ is the ideal gas constant and

$T$ is the ambient temperature.

Due to the complex mixing states of ambient aerosol, we divided aerosol particles into two modes

with respect to their hygroscopicity: a less-hygroscopic mode (LH, $\kappa <= 0.2$) and a more-hygroscopic

mode (MH, $\kappa > 0.2$) (Shi et al., 2022).

### 2.3 Other measurements

In this study, the particle number size distribution (8-750 nm) was measured using a Scanning

Mobility Particle Sizer (SMPS, Model 3080, TSI Inc., USA). Water-soluble inorganic ions (Na$^+$, NH$_4^+$,

NO$_3^-$, SO$_4^{2-}$) were quantified by a URG-9000 Ambient Ion Monitor (Thermo Fisher, USA) through a

PM$_{2.5}$ sharp-cut cyclone inlet, while the mass concentration of elemental carbon (EC) and organic carbon

(OC) were determined using an OC/EC aerosol analyzer (Sunset Laboratory, Forest Grove, OR) (Bae et

al., 2004).

In addition to online measurements of the particle chemical composition, offline samples were

collected twice daily (07:00–19:00 and 19:00–07:00 the following day) on quartz fiber filters using a

high-volume air sampler (TISCH, TE6070DV-BL) operating at a flow rate of 1.13 m$^3$·min$^{-1}$. All collected

samples were wrapped in foil and stored at ~ 4 °C until analysis. Field blank samples were also collected

before and after the sampling by mounting a prebaked blank filter onto the sampler for about 10 min

without sucking any air. The major chemical components of PM$_{2.5}$, including major and trace elements

as well as water-soluble ions, were subsequently analyzed using an energy-dispersive X-ray fluorescence

spectrometer (ED-XRF, Epsilon 4, Malvern Panalytical, Netherlands) and ion chromatography (940

Professional IC Vario, Metrohm). Detailed information for the offline chemical analysis can be found in

(Feng et al., 2023).

Meteorological conditions, including wind direction, wind speed, ambient temperature, and RH,

were continuously measured from the Mt. Hua meteorological station. All the instruments for aerosol

measurements were placed in an air-conditioned room, where temperature was maintained at 25 °C. The

aerosol sampling inlet was located on both sides of the room. The aerosol was sampled via a low-flow

PM$_{2.5}$ cyclone inlet, passed through a Nafion dryer, and directed to different instruments through stainless



steel or conductive black tubing using an isokinetic flow splitter. The sampling air was dried to RH below

20 % using a Nafion diffusion dryer.

### 2.4    Analysis of air mass origins

To investigate the origins of air masses reaching our observational site, the Hybrid Single Particle

Lagrangian Integrated Trajectory (HYSPLIT) transport and dispersion model (Draxler and Hess, 1998)

was employed. In the calculations, 24-hour backwards trajectories were computed for air parcels arriving

at the sampling location (2060 m a.s.l) with a temporal resolution of 2 hours.

## 3    Results and discussion

### 3.1    Overview

Figure 1 illustrates the time series of wind direction, wind speed, RH, ambient temperature, $PM_{2.5}$

concentrations, and black carbon (BC) mass concentrations throughout the entire experimental period.

Overall, the average wind speed during the campaign was around $5.0 \pm 3.2$ m/s, with the prevailing wind

directions originating from the west and northwest. The $PM_{2.5}$ mass concentration was generally below

20 $\mu$g/m³, with an average value of $8.2 \pm 6.7$ $\mu$g/m³, aligning with the winter 2020 averages observed at

the Mt. Hua site reported by Feng et al. (2023) (Feng et al., 2023). In contrast to the heavily polluted

Guanzhong Basin urban areas, the average $PM_{2.5}$ levels were notably higher (e.g., $68.0 \pm 42.8$ $\mu$g/m³ in

Xi'an; Chen et al., 2021). This discrepancy, which is consistent with previous studies (Dai et al., 2018),

indicates a comparatively reduced influence of ground-level anthropogenic emissions at the current site,

plausibly attributed to particle deposition or dilution during the transport from lower altitudes to the

mountaintop (Feng et al., 2023). From Fig. 1, it is clearly to observe that meteorological conditions

substantially changed after November 6, characterized by a decrease in temperature, a slight increase in

wind speed, and a shift in the wind direction from west to northwest. Due to the significant temperature

drop during this period, domestic heating in this area was initiated. The notable differences in the

meteorological conditions as well as anthropogenic emissions during the two distinct periods can result

in different atmospheric processes and transport of aerosols. This would in turn lead to variations in the

aerosol chemical composition as well as their physicochemical properties, in particular, aerosol

hygroscopicity. Detailed analysis of aerosol hygroscopicity and their chemical composition, taking into



account of the influence of air mass origins and domestic heating activities, for the two periods will be further analyzed in Sect. 3.3.

Fig. 2 (a–e) shows the temporal evolution of the $\kappa$-PDF for different particle sizes, with the black line representing the average $\kappa$. The results reveal that the average $\kappa$ for particles at 30, 60, 100, 150, and 200 nm were 0.20, 0.21, 0.25, 0.27, and 0.30, respectively. Chen et al. (2021) obtained an average $\kappa$ value of $0.14 \pm 0.04$ for the water-soluble components of 100 nm particles at the ground-level environment of the Guanzhong Basin during winter based on particle chemical composition measurements. Considering the contribution of water-insoluble components, the overall hygroscopicity would be even lower. We observed that at our observational site, water-soluble inorganic components were the dominant composition in particles (average of 77 % in mass fraction, see Fig. 2e), while the organic components contributed the most (54 % in mass fraction) at the ground-level Guanzhong Basin background site. Though bulk-phase chemical composition may deviate from that of size-resolves ones, this may partially explain the large difference in aerosol hygroscopicity between the two sites at the same region. Particles sampled at the ground level indicating the distinct sources, secondary processes for atmospheric aerosols in the free troposphere compared to ground-level environments.

The averaged $\kappa$-PDF in Fig. 3a reveals that particles of different sizes at the current study were dominated by the MH mode, indicating the aerosols at the site had a mild degree of external mixing of different sources. This characteristic is similar to the results from other high-altitude regions, which could be explained by the slight influence of the advection of aerosols from the PBL (Dingenen et al., 2005; Holmgren et al., 2014; Sjogren et al., 2008). As illustrated in Figure 3a, the contribution of the MH mode becomes more pronounced as particle size increases, which means nucleation and Aitken mode particles (diameter at 30 and 60 nm) tend to contain higher number fraction of LH mode particles (see Table 1). Nucleation and Aitken mode particles are typically from new particle formation (NPF) and subsequent growth or anthropogenic sources injected from the PBL. However, NPF events were rarely observed (only once) at the current site during this study, suggesting anthropogenic emissions within the PBL were the primary sources for these smaller, less hygroscopic particles. This on the other hand indicates that accumulation mode particles (e.g., those with diameters at 100-200 nm) at the site, characterized by a higher degree of internal mixing, may stem from a different origin, being most likely long-range transported along with extensive aging processes.



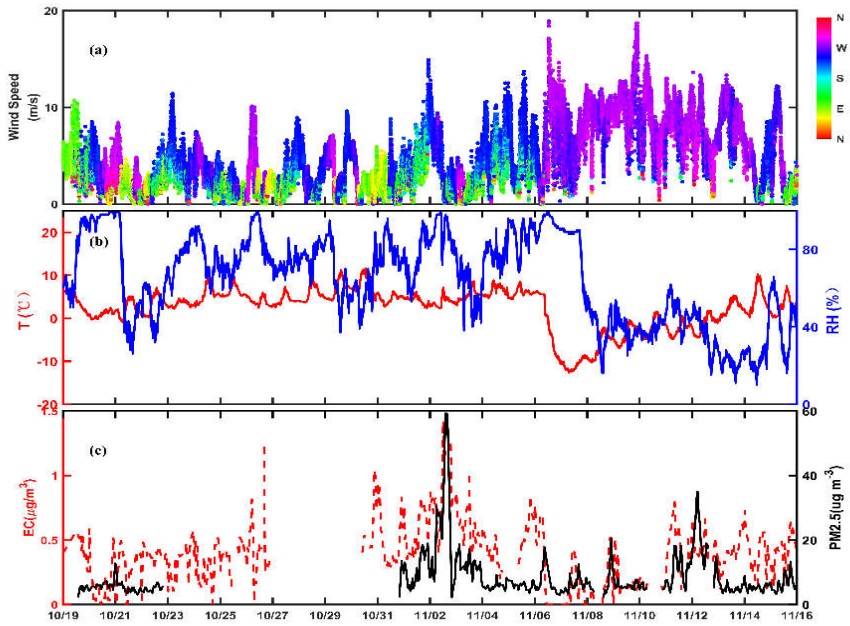


**Figure 1. Meteorological conditions over the whole observation period. (a) wind speed, wind direction; (b) the temperature and RH; (c) PM$_{2.5}$ and EC mass concentrations.**



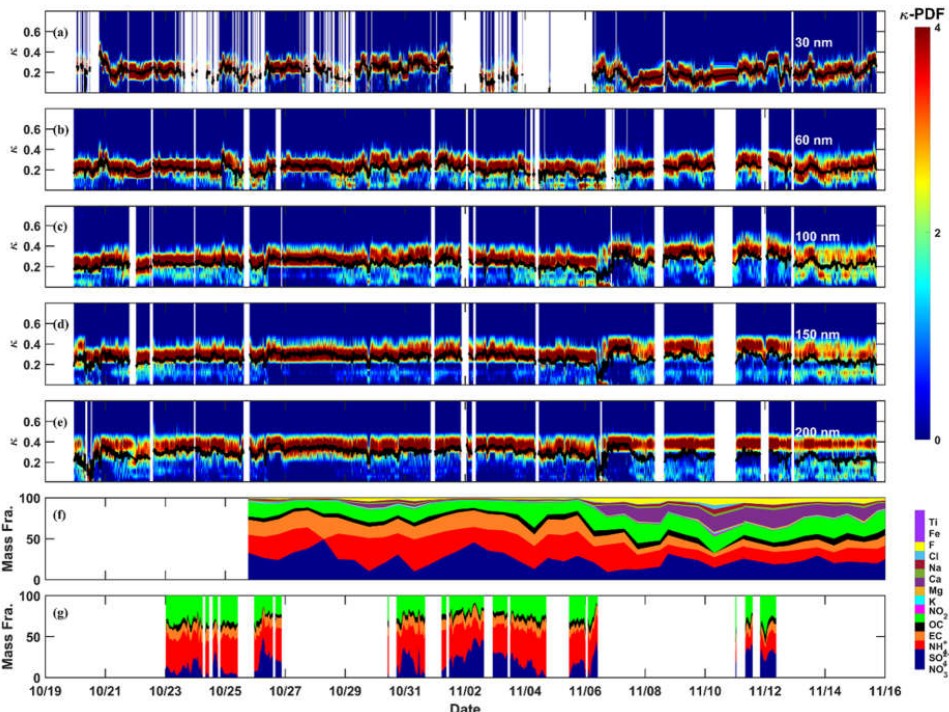

Figure 2. (a-e) Time series of $\kappa$-PDF for different particle sizes (with the black line indicating the mean $\kappa$ value) and (f-g) time series of chemical composition derived from offline measurements and URG measurement, respectively, during the sampling period.








**Table 1 Statistics of hygroscopicity parameter $\kappa$ for particles of different sizes.**

| Diameter | 30 nm | 60 nm | 100 nm | 150 nm | 200 nm |
|---|---|---|---|---|---|
| **All groups** | | | | | |
| $\kappa$ | 0.20 | 0.21 | 0.25 | 0.27 | 0.30 |
| Std. dev | 0.06 | 0.04 | 0.05 | 0.04 | 0.05 |
| **More-hygroscopic** | | | | | |
| $\kappa$ | 0.27 | 0.27 | 0.30 | 0.32 | 0.35 |
| Std. dev | 0.03 | 0.02 | 0.03 | 0.03 | 0.03 |
| NF | 0.58 | 0.63 | 0.74 | 0.82 | 0.84 |
| Std. dev | 0.28 | 0.17 | 0.14 | 0.12 | 0.13 |
| **Less-hygroscopic** | | | | | |
| $\kappa$ | 0.13 | 0.13 | 0.11 | 0.11 | 0.11 |
| Std. dev | 0.04 | 0.03 | 0.03 | 0.03 | 0.04 |
| NF | 0.40 | 0.35 | 0.25 | 0.17 | 0.16 |
| Std. dev | 0.28 | 0.17 | 0.13 | 0.11 | 0.13 |

### 3.2 Diurnal variations of $\kappa$-PDFs

No clear diurnal variation was observed in the $\kappa$-PDF of particles at most sizes at the current site, except for a minor broadening in the $\kappa$-PDF during daytime for 30-nm particles, as depicted in Fig. 3b-f. The relatively constant $\kappa$-PDF of particles at most sizes throughout the entire day is consistent with the

stable diel variation in the number fraction of each hygroscopic mode and their respective hygroscopicity (see Fig. S1). The wider $\kappa$-PDF for 30 nm particles during the daytime was mainly attributed to the elevated number fraction of particles in the less-hygroscopic mode. Given that our observational site is consistently located within the residual layer or the free troposphere throughout the day, the influence of vertical diffusion of pollutants from the PBL, being primarily anthropogenic, was expected to be minimal.

This is particularly obvious for larger particles, which were abundant at the site, while for smaller particles, this influence became more notable due to their lower concentration that a slight vertical upward transport from the PBL may potentially alter their $\kappa$-PDF characteristics.

As aerosol hygroscopicity is ultimately determined by their chemical composition, their diurnal pattern observed aligns neatly with the daily trend in the mass fractions of each component within the

particles, as shown in Fig. S2. Note that the chemical composition data from the URG were used in this analysis, as the offline measurements, while having better temporal coverage, were limited to half-day resolution. Interestingly, notable differences were observed in meteorological variables, including



ambient temperature and RH, as well as the concentration of atmospheric trace gases like $O_3$ and $SO_2$, between daytime and nighttime at the site (see Fig. S3 and Fig. S4). Thus, it is reasonable to expect that,

during daytime, under the conditions of higher solar radiation, elevated temperatures, and increased levels of atmospheric oxidants, various secondary species are likely to form, probably through photochemical oxidation processes. These newly-formed species will subsequently partition or condense onto pre-existing particles, thereby altering their chemical composition and, consequently, their hygroscopicity. However, such an influence on aerosol chemical composition and their hygroscopicity

was not obviously observed at the current study. A plausible explanation for this could be the insignificant yield of these secondary compounds, potentially due to the scarcity of their atmospheric precursors during this season. These results further confirm that the current aerosols, particularly larger particles, were primarily originated from regional or long-range transport with longer atmospheric aging, with limited contributions from local lower altitude sources or in situ secondary formation.


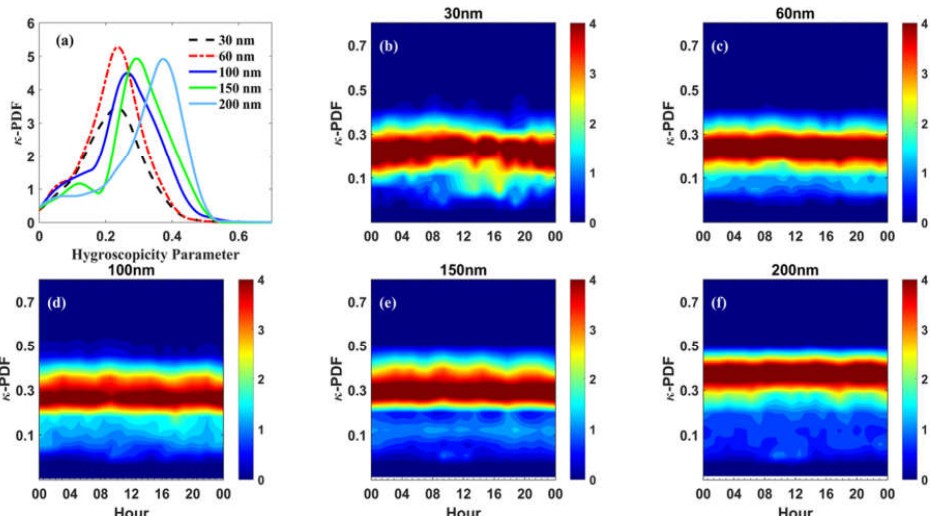

**Figure 3. (a) Average $\kappa$-PDF for particles at different sizes; (b-f) diurnal variations of the $\kappa$-PDF for particles of different sizes measured during the campaign.**





### 3.3 The impact of regional emissions and long-range transport on aerosol hygroscopicity of high-altitude aerosols

Despite the absence of a clear diurnal pattern in the particle chemical composition by online analysis, semi-diurnal offline measurements demonstrated significant compositional changes between the first and second half of the campaign. Specifically, starting from November 6, the contributions of $Ca^{2+}$ and $Fe^{2+}$ to $PM_{2.5}$ mass increased substantially, accounting for up to approximately 20 % of the total particle mass, indicating a shift in aerosol sources after this time. Given that $Ca^{2+}$ and $Fe^{2+}$ are typically recognized as mineral dust tracers (Feng et al., 2023; Kouyoumdjian and Saliba, 2006), these observations strongly suggest an episode of distinct long-range transport of mineral dust after November 6. This signature, being consistent with previous findings (Liu et al., 2024), was further confirmed by the backward trajectory analysis as shown in Fig. 4. The results revealed that air masses arriving at the current site during this time were mainly originated from the northwest, passing through the Mongolia-Inner Mongolia corridor and Tengger Desert region at ~1400 m altitude before reaching our site (Cluster 3 and 5). On the other hand, this period also coincided with the initiation of regional domestic heating, creating a unique scenario where long-range transported dust components mixed with regional emitted anthropogenic pollutants. Consequently, the arriving air masses, enriched with both mineral dust and heating-related emissions, likely modified the background aerosol chemical composition at our observational location (Du et al., 2022).

To comprehensively evaluate the influence of long-range transport and regional emissions on aerosol hygroscopic properties, we compared the size-resolved aerosol hygroscopicity parameter $\kappa$ among six trajectory-identified air mass clusters (Fig.5a). During the periods without significant influence of mineral dust and prior to the heating activities (i.e., Cluster 1, 2, 4 and 6), aerosols particles showed a clear size-dependency of $\kappa$, with comparable $\kappa$ values observed for particles at the same sizes. Air masses associated with these clusters, particularly Cluster 1 and 4, primarily transported over relatively short distances from southeastern and southwestern regions. These trajectories passed through the heavily polluted Guanzhong Basin urban agglomeration, potentially reflecting the atmospheric characteristics of this region. In contrast, aerosol particles in Cluster 3 and 5 displayed relatively constant $\kappa$ across most particle sizes, except for particles smaller than 100 nm, which probably had local origins rather than long-range transport, as previously discussed.





In Cluster 5, 200 nm particles showed the lowest $\kappa$ value (~0.25) compared to other clusters (0.3).

This reduction in overall hygroscopicity is attributed to the decreased hygroscopicity ($\kappa \approx 0.08$) and elevated number fraction (23 %) of LH mode particles, in contrast to other clusters where LH particles constituted 15% in numbers with $\kappa \approx 0.12$. Consistent with expectation, the increased fraction of LH mode particles may partly originate from mineral dust, as they were typically hydrophobic or weakly hygroscopic. This mineral dust influence was further confirmed by the strong correlation between $SO_4^{2-}$

with $Ca^{2+}$ ($R^2 = 0.83$) in Cluster 5, contrasting with their weak associations ($R^2 \approx 0.1$) during other clusters, where $SO_4^{2-}$ instead well linked to $NH_4^+$ ($R^2 \approx 0.78$). The co-variation of $SO_4^{2-}$ and $Ca^{2+}$ indicates that they possibly shared the same origins (Sullivan et al., 2009), and $Ca^{2+}$ may primarily exist in the form of hydrophobic $CaSO_4$ ($\kappa \approx 0.01\text{-}0.05$) during this episode, reinforcing the observed hygroscopicity decline. Furthermore, domestic heating activities were a significant source of primary organic aerosol particles,

which exhibited weak hygroscopicity as suggested by previous studies (Shi et al., 2022) and may also contribute to the elevated proportion of LH mode particles during this episode. Though no direct source apportionment of organic aerosols can be obtained by the current study, a notably higher fraction of organic aerosols was observed in $PM_{2.5}$ of Cluster 5 compared to other clusters. We hypothesize that the elevated fraction of organic aerosols in Cluster 5 could be primarily driven by domestic heating emissions,

consistent with previous study (Du et al., 2020), which reported a substantial rise in the fraction of organic aerosols following the onset of domestic heating. Additionally, Cluster 5 exhibited a slight increase in the overall aerosol hygroscopicity as particle size decreases, which can be explained by the enhanced hygroscopicity of LH mode particles together with a modest reduction in their number fraction. Given that mineral dust mainly resided in large particles (e.g., > 100~200 nm), this observed size-dependent

hygroscopicity trend suggests a reduced contribution of mineral dust to the hygroscopicity of smaller particles in Cluster 5.

Despite both being influenced by dust events and domestic heating activities, aerosol particles in Cluster 3 exhibited higher hygroscopic growth, mainly due to their larger number fraction of MH mode particles, accompanied by the higher hygroscopicity of LH mode particles. Interestingly, striking high

RH levels (around 80 %) were observed in Cluster 3, nearly doubled those observed in Cluster 5. Such high-RH conditions may promote the aging of aerosols particles through processes, such as multi-phase or aqueous phase reactions, potentially altering their chemical composition and may explain their elevated hygroscopicity (Tong et al., 2020). This hypothesis aligns with the results of Du et al. (2022),





who observed that the contribution of aqueous-formed water soluble oxidized organic aerosols to the

total water-soluble organic aerosols increased from 11.21 % under low-RH levels to over 40 % at RH >

80 %, indicating a significant transformation in the organic aerosol composition. On the other hand, we

noticed that the average $\kappa$ of MH mode particles in Cluster 3 (also in Cluster 5) was markedly greater

relative to other clusters. Building upon our previous reasoning, we hypothesize that the elevated RH

levels in Cluster 3 facilitated multi-phase or aqueous phase reactions, which not only enhanced the

hygroscopicity of LH mode particles, but also generated substantial amount of highly hygroscopic

materials. These aerosol constituents may have persisted until Cluster 5, contributing to its observed

aerosol hygroscopic properties. However, without detailed compositional analysis at molecular level, the

exact mechanisms responsible for this exceptionally high aerosol hygroscopicity remained unclear.

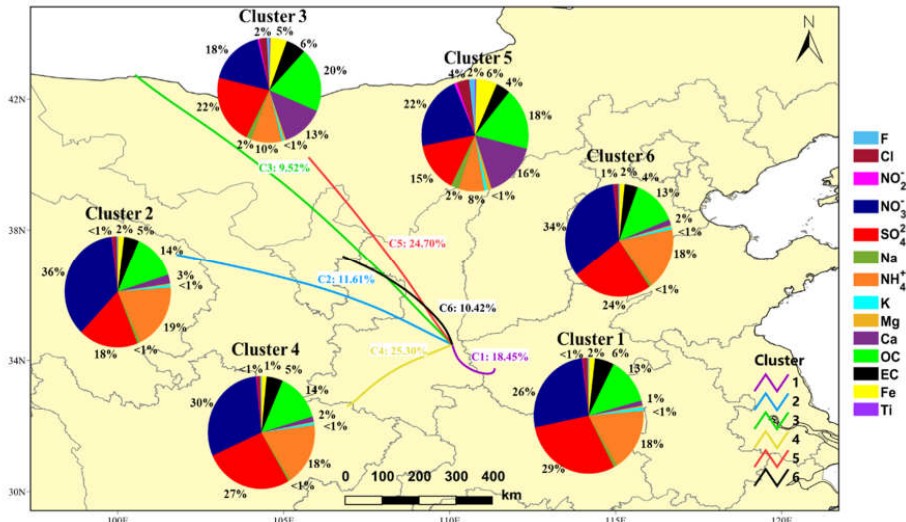

**Figure 4. Cluster analysis of 24 h backward trajectories at 2060 m above ground level at the sampling**
**site and chemical composition of aerosols during the six trajectory-identified clusters.**





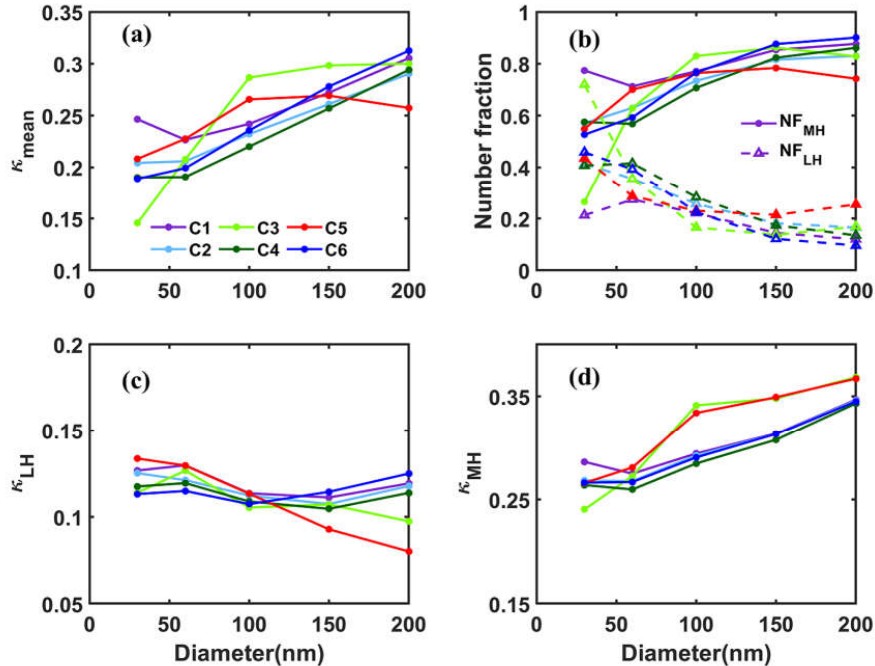

**Figure 5. Cluster analysis corresponding to (a) mean aerosol hygroscopicity parameters $\kappa$, (b) the number fraction of LH and MH mode particles, (c) the mean $\kappa$ values of LH and MH mode particles at different sizes.**





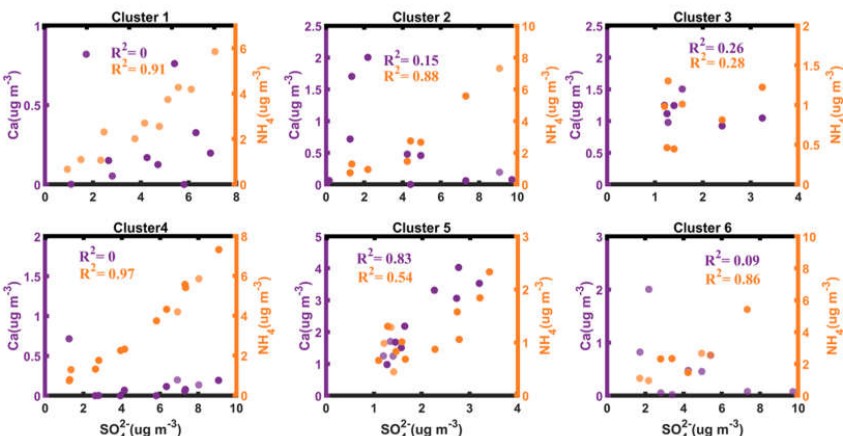

**Figure 6. The correlation between SO4²⁻ and Ca²⁺, and SO4²⁻ and NH4⁺ in PM2.5 for different air masses.**

### 3.4 Comparison to other free atmosphere ambient measurements

The unique environment of free tropospheric observation sites, being distant from anthropogenic pollution sources, provides a more representative characterization of regional-scale aerosol properties. However, current measurements of aerosol hygroscopicity in the global free troposphere are remarkably sparse (Fig. 7). These available datasets shows that hygroscopic parameter $\kappa$ for free tropospheric aerosols typically ranged from 0.15 to 0.35, with ours falling within the median of reported values.

Winter-time aerosols at Jungfraujoch (3580 m) revealed comparable hygroscopicity to our results, while their higher annual mean ($\kappa \approx 0.3$) implies substantial contribution from secondary inorganic components or more aged components during other seasons (Kammermann et al., 2010). The Monte Cimone Observatory (Italy), situated in the transition zone between the boundary layer and free troposphere, exhibited slightly reduced aerosol hygroscopicity, likely due to vertical transport of less hygroscopic

pollutants, which were less-aged, from the boundary layer (Dingenen et al., 2005). Similarly, at lower-altitude sites, such as Huangshan (1865 m) and Puy de Dôme (1465 m), aerosol hygroscopicity was further reduced as a result of enhanced anthropogenic influence (Asmi et al., 2012; Wu et al., 2018a). These results highlight an altitude gradient in aerosol characteristics, where the influence of boundary layer decreased with elevation, resulting in progressively more aged and more hygroscopic aerosols at




higher altitudes compared to fresher, less aged and less hygroscopic ones close to the surface. The Izaña

Atmospheric Observatory, located in the northeastern Atlantic at 2370 m altitude, provided a different

case of marine-influenced aerosol properties (Swietlicki et al., 2000). Their measurements revealed

exceptionally high hygroscopicity for 50 nm particles ($\kappa \approx 0.36$), mostly likely stemming from the

dominance of inorganic sea salts in the studied particles, contrasting markedly with continental aerosols

from tropospheric sites. These comparative analyses demonstrate fundamental differences in atmospheric

processes and interactions between free tropospheric and boundary layer environments by distinguishing

natural background aerosol properties from anthropogenic influences, these works may establish critical

benchmarks for modeling aerosol-climate interactions.

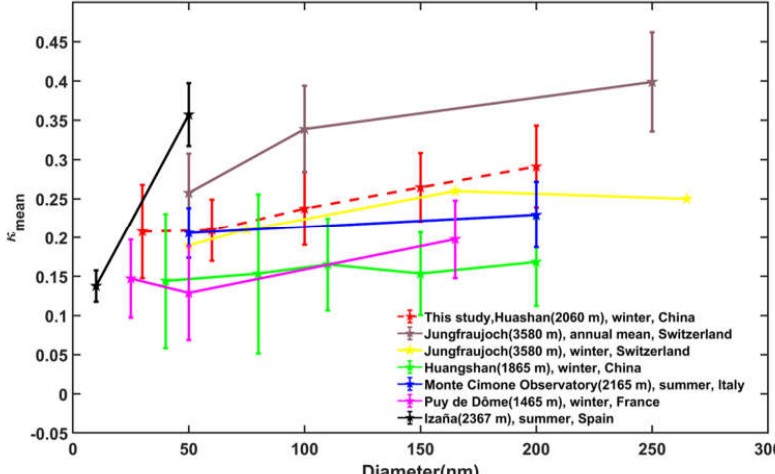


**Figure 7. Comparison of aerosol hygroscopicity measured at different high-altitude sites around the world.**

### 4. Conclusion and implications

This study advances our understanding of aerosol hygroscopicity and mixing states in the lower free

troposphere over central China through comprehensive measurements at Mt. Hua (~2060 m a.s.l.). The

observed size-dependent hygroscopic growth pattern and externally mixed state of aerosols provide

direct evidence of particle transformation during transport in the free troposphere. The absence of diurnal

variations in aerosol hygroscopicity confirms minimal influences from boundary layer dynamics, making

this site ideal for representing background atmospheric conditions. Particularly noteworthy is the RH-



driven hygroscopic enhancement of mineral dust particles, suggesting substantial chemical processes previously underestimated in such environments. These distinct aerosol characteristics highlight the unique atmospheric processes occurring where anthropogenic, natural, and transported air masses interact.

The implications extend beyond investigating long-range transport and free troposphere-boundary layer interactions. Alpine aerosols like those at Mt. Hua act as critical indicators of regional atmospheric properties. Our comparative analysis of mass extinction efficiency (MEE) - a key parameter describing aerosol light attenuation per unit mass, revealed striking altitude dependence: free tropospheric aerosols at the summit exhibited MEE values of 4.29 $m^2$ $g^{-1}$, which was three times greater than surface-level

observations (1.30 $m^2$ $g^{-1}$), despite much lower particle mass aloft. The discrepancy mainly arises from enhanced hygroscopic growth of aged free tropospheric aerosols, which have experienced longer atmospheric processing with reduced influences from local emissions. Such pronounced vertical variations in MEE highlight the necessity of incorporating regional-scale or altitude-resolved hygroscopicity into climate models, particularly for humid or high-altitude settings where such effects

may dominate aerosol radiative forcing.



**Data availability.**

The data of this paper can be obtained from https://doi.org/10.5281/zenodo.15589884 (Shi et al., 2025).

**Author contributions.**

JH, NM, QW, and LL conceptualized and supervised this study. JS, ZZ, LL YZ,SH, SZ, ML, LX, WR, and JT conducted the field campaign. JS and LL conducted the data analysis. HX contributed to the instrument maintenance. JH, JS, LL, ZZ, NM, JT, YZ, QW discussed the results. JS drew the plots and JH and JS wrote the draft. JH, JS and AW proofed the paper and edited the paper with contributions from 400 all co-authors.

**Competing interests.**

The authors declare that none of the authors has any competing interests.

**Acknowledgments.**

This research was supported by the Guangdong Basic and Applied Basic Research Foundation (Grant 2024A1515510020); the "Western Light"-Key Laboratory Cooperative Research Cross-Team Project of Chinese Academy of Sciences (xbzg-zdsys-202219); the Science and Technology Program of Guangdong (2024B1212080002); the National Natural Science Foundation of China (grant no.42175117, 410 42375072); the Guangzhou basic and applied basic research project-Guangzhou Science and Information Bureau project (grant no. 2025A04J3520).




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
