# Peer review of "Measurement Report: Hygroscopicity and mixing state of submicron aerosols in the lower free troposphere over central China: local, regional and long-range transport influences"

_EGUsphere, 2025_

## Author Comment (AC1)

**Response to reviewer #1**

Shi and Zhang et al.'s manuscript presents hygroscopicity and mixing state of submicron aerosols at several mobility diameters at Mt. Hua during fall 2021 using HTDMA system. This study presents a valuable and comprehensive dataset on aerosol hygroscopicity in the lower free troposphere over China, supported by concurrent aerosol chemical composition measurements. The manuscript primarily serves as a measurement report based on a robust dataset, rather than providing extensive interpretation or discussion. Given the scientific merit and relevance of the dataset, I believe this manuscript can be considered for publication in Atmospheric Chemistry and Physics after the authors adequately address the following comments and revise accordingly. Most of all, authors are encouraged to related aerosol hygroscopicity, chemical composition, and airmass origins or meteorology impacting aerosol process in the atmosphere.

We greatly appreciate referee#1's positive feedback and constructive suggestions which are of great value for improving the quality of our paper. Below are our point-to-point responses to the referee's comments.

1. L114: The parameter "A" is defined with an equation; however, a clearer explanation of its physical meaning and implications is necessary.

Response:

Thank you for your comment. We realize that the explanation provided in the original manuscript may not have been clear. We have revised the entire formula based on $\kappa$-Köhler theory. The revised formula is as follows:

$$\kappa = (GF^3 - 1)\left(1 - \frac{RH}{K_e}K_e\right),$$

$$K_e = \exp\left(\frac{4\sigma_{s/a}M_w}{RT\rho_w D_{dry}}\right),$$

GF is the hygroscopic growth factor measured by HTDMA at 90 % RH. $D_{dry}$ is defined as the particle diameter selected by the first DMA under dry conditions (RH < 10 %) at 25 °C. $\rho_w$ and $M_w$ are the density and molecular weight of water. $\sigma_{s/a}$ is the surface tension of the droplets, which is assumed to be that of pure water ($\sigma_{s/a}$ = 0.0728N m$^{-2}$). $R$ is the ideal gas constant and $T$ is the ambient temperature, $K_e$ is the Kelvin correction factor term.

2. L115: "Dry" diameter appears to refer to a particle size at 20°C, as suggested in Section 3. Please clarify this here and provide a more detailed explanation.

Response:

Thanks for your specific comment. We have revised the sentence in Line 125 for clarity. The updated sentence now reads: "$D_{dry}$ is defined as the particle diameter selected by the first DMA under dry conditions (RH < 10 %) at 25 °C.".

3. L173: The term "$\kappa$-PDF" is introduced without definition. Please define it clearly upon first mention.

Response:

Thank you for this constructive suggestion. A clear definition of "$\kappa$-PDF" has been included at its first appearance in of the revised manuscript, as follows: "The $\kappa$ probability density function ($\kappa$-PDF),

indicative of a statistical distribution that describes the variation in hygroscopicity among an aerosol population, was derived from the GF probability density function (GF-PDF), which was retrieved from the measured GF distribution function (GF-MDF) using the TDMAinv algorithm (Gysel et al., 2009).".

4. L197: Aerosols with diameters of 100–200 nm in number concentration typically represent secondary aerosols such as sulfate and nitrate, which often show mass size distribution peaks at 400–500 nm in previously published studies of AMS measurements. Could you provide the likely chemical composition of particles in the 100–200 nm range based on your measurements or referring to previous studies?

Response:

Thank you for this constructive suggestion. To elucidate the likely chemical composition of particles in the 100–200 nm range, we have incorporated a discussion referencing previous studies at Line 222 of the revised manuscript: "This finding on the other hand suggests that the larger particles (e.g., 100-200 nm) at the site, characterized by a higher degree of internal mixing, may stem from different origins and are most likely subject to long-range transport with extensive aging processes. This interpretation is supported in part by Li et al., (2011), who conducted measurements during similar seasons at Mt. Hua and reported that particles in the 100–400 nm range were predominantly composed of secondary inorganic species, such as $SO_4^{2-}$, $NO_3^-$, and $NH_4^+$, which typically have experienced longer aging processes."

5. Figure 2: Panels (a) to (e) show time series for particles from 30 nm to 200 nm, but the text within the panels is too small to read comfortably. Please enlarge the text for improved readability. Additionally, the term "URG" in the caption (and in line 235) might be better replaced with a descriptor indicating continuous or online measurement.

Response:

Thank you for this constructive suggestion. We have increased the font size of the text in all sub-panels of Figure 2 to enhance readability. In addition, we have replaced "URG" with "online measurement" in both the figure caption and in the revised manuscript to provide a clearer description of the data source.

[Figure]

**Figure 1. (a-e) Time series of $\kappa$-PDF for different particle sizes (with the black line indicating the mean $\kappa$ value) and (f-g) Time series of chemical composition of PM2.5 obtained using offline and continuous online measurements, respectively, during the sampling period.**

6. Figure 2 (continued): Starting from November 14, a decrease in hygroscopicity (k) coincides with an increase in the mass fraction of elemental carbon (EC), particularly for particles in the 60 – 150 nm diameter range. Was this shift possibly associated with the advection of primary particles from urban areas? This period may serve as a useful contrast case relative to periods of higher hygroscopicity.

Response:

Thanks for this very constructive suggestion. In the original manuscript, we had discussed the influence of primary aerosols from domestic heating on the hygroscopicity during this period. However, we acknowledge that our description might have been too weak to clearly highlight this point. In response to your valuable comment, we have revised the entire paragraph as follows: "As noted earlier, beginning on November 6, which encompassed the entire duration of Cluster 3, 5, 1 DH, 2 DH, and 4 DH, regional domestic heating was initiated, which may emit substantial amounts of primary aerosols, such as black carbon (BC) and primary organic aerosols (POA). These aerosols were likely transported to our observational site via advection. This interpretation aligns with the findings of Du et al. (2022), who reported a marked increase in the fraction of organic aerosols as well as BC at Mt. Hua following the initiation of domestic heating. Though no direct source apportionment of organic aerosols can be

obtained by the current study, a moderate increase (approximately 7 %) in the organic mass fraction in PM$_{2.5}$ during Cluster 3 and 5, followed by a more pronounced rise in both organic (10 %) and BC fractions (5 %) during Cluster 1 DH, 2 DH, and 4 DH was observed compared to other clusters (see Fig.6), further supporting our previous hypothesis. Thus, the elevated levels of these primary aerosols, typically exhibited weak hygroscopicity (Shi et al., 2022), coupled with the high contents of weakly hygroscopic mineral dust, may synergistically drive the continued decline in aerosol hygroscopicity throughout this period."

7. Figures 4 and 6: Please consider enlarging the text in both figures to improve readability.

Response:

Thanks for your suggestion. We have enlarged the text in Figures 4 and 7 (formerly Figure 6) in the revised manuscript to improve readability. And the enlarged Figures are shown below.

[Figure]

**Figure 2. Cluster analysis of 72 h backward trajectories at 2060 m above ground level at the sampling site during the five trajectory-identified clusters. The line colors denote different clusters, i.e., purple for Cluster 1, yellow for Cluster 2, green for Cluster 3. black for Cluster 4, and blue for Cluster 5. (Figure 2 in manuscript)**

[Figure]

**Figure 3. Proportions of chemical composition in different clusters. (Figure 6 in manuscript)**

[Figure]

**Figure 4. The correlation between SO4²⁻ and Ca²⁺, and SO4²⁻ and NH4⁺ in PM2.5 for different air masses. (Figure 7 in manuscript)**

8. Figures 4 and 5: The six clusters identified in these figures appear to form two broad groups: clusters C3 and C5 versus the remaining clusters. Back-trajectory analysis in Figure 4 suggests that C3 and C5 are associated with air masses from cold and dry regions. In terms of k values, these clusters exhibit bimodal distributions, with lower values in the low hygroscopicity (LH) mode and higher values in the

moderate hygroscopicity (MH) mode for particles larger than 100 nm. Could the authors discuss how aerosol composition and hygroscopicity differ across these clusters?

Response: Thanks for the comments. We agree with the reviewer that the connection between aerosol composition and hygroscopicity is highly essential. Therefore, we have updated the trajectory analysis using 72-hour trajectories and made more discussions to relate aerosol composition with hygroscopicity across different clusters. The relevant part in Sect. 3.3 was revised as: "
[revised manuscript text omitted]

9. I am not sure how this trajectory analysis can support aerosol hygroscopicity measurements, also as 6 clusters of air mass trajectories are not clearly distinguished and related to the aerosol measurements. Do the authors consider surface emissions based on trajectory height or microscale/synoptic meteorology

influencing aerosol hygroscopicity during transport?  While I understand the manuscript is intended primarily as a measurement report, a brief discussion or summary connecting air mass origin, composition, and hygroscopic behavior would significantly enhance the interpretation of these results.

Response:

Thank you for your suggestion. We understand the reviewer's concern that the identified clusters of air mass trajectories may not be closely related to the aerosol measurements. As the reviewer previously noted, correlating aerosol hygroscopicity with their chemical composition would provide a more direct approach to evaluate the influence of different aerosol sources on their physicochemical properties. In response, we have revised Sect. 3.3 accordingly to improve the clarity and representation of these relationships. Additionally, we have included the vertical height profiles of the five air mass trajectories (see Fig.S6) and the time series of the boundary height at the site throughout the measurement period (see Fig.S7). The results indicate that all trajectories remained well above the boundary layer, suggesting that the observed air masses were not significantly influenced by local surface emissions. However, it is worth noting that these air masses passed through regions characterized by strong vertical mixing, such as those influenced by valley breeze, which could potentially uplift surface pollutants to the observational sites. At present, distinguishing such contributions remains challenging within our analysis.

As suggested by the reviewer, we have incorporated a brief summary at the end of Sect. 3.3 to better connect air mass origin, composition, and hygroscopic behavior to enhance the interpretation of the results. The summary is also provided below as: "In summary, during the first half of the campaign, when air masses mainly passed through the heavily polluted Guanzhong Plain urban agglomeration, aerosols were primarily composed of secondary inorganic species and exhibited the highest hygroscopicity. Starting around November 6, increased influences from both mineral dust and domestic heating activities led to a noticeable decline in aerosol hygroscopicity, particularly in larger particles. This reduction was largely attributed to the rising levels of weakly hygroscopic components, such as mineral dust tracers (e.g., $Ca^{2+}$, $Fe^{2+}$), organic components as well as BC, highlighting the combined effects of long-range transport and regional emissions on aerosol composition and properties.".

[Figure]

Figure 5. The diurnal variations in temperature and humidity for Cluster 3 and Cluster 5 during the

[Figure]

**Figure 6. 72 h backward trajectory clusters at 2060 m above ground level. Different colors represent trajectory altitudes in distinct ranges. Percentages indicate the proportion of each cluster.**

[Figure]

**Figure 7. Time series of boundary layer height variations during the observation period.**

---

## Author Comment (AC2)

**Response to reviewer #2**

The manuscript by Shi and Zhang et al. presents a measurement report, providing detailed description of the hygroscopicity and mixing state of the aerosols in mountain Hua in central China for few months in 2021. Despite focusing on a single location and relatively short time period, the manuscript is worth publishing due to the measurement location, and thus certain site characteristics, which have not been widely reported. The manuscript is overall well written and provides all necessary details, but I have minor concerns and suggestions for revisions that should be considered before accepting for publication.

We greatly appreciate referee#2's positive feedback and constructive suggestions which are of great value for improving the quality of our paper. Below are our point-to-point responses to the referee's comments.

1. Regarding the trajectory analysis, I wonder why the authors have selected the 24-hour long trajectories. I think reasoning for this selection is crucial and should be added to the text, especially when considering the fact that aerosols, especially the larger ones measured at the site, could originate very far. In many other studies utilizing backward trajectory analysis, longer trajectories (to my knowledge, often at least 48 hours) are usually employed (see for example, Räty et al., 2023; Khadir et al., 2023; Xu et al., 2021) to make sure long-range transport is properly captured.

Response:

We appreciate this very constructive suggestion. Following the reviewer's suggestion, we have recalculated the backward trajectories using a 72-hour duration. The resulting clusters obtained from 72-hour trajectory analysis is presented in Fig.1 below. As the results differ slightly from the previous version, we have updated the results and discussion in Sect. 3.3 accordingly. The revised content is provided below for reference: "
[revised manuscript text omitted]

[Figure]

**Figure 1. 24 h backward trajectories (left) and 72 h backward trajectories (right) at 2060 m above ground level during the sampling period.**

2. Line 27: "improving predictions…climate impacts" Please reword as the current structuring is unclear.
Response:
Thanks for your specific comment. We have revised the sentence in Line 26 for clarity. The updated sentence now reads: "Understanding the hygroscopicity and mixing state of atmospheric aerosol particles is crucial for accurately assessing their role in cloud formation and subsequent climate impacts.".

3. Line 34: I assume the sizes of the particles are in the brackets, please make it clear by stating it explicitly (e.g., dp = 30 nm).
Response:
Thank you for this constructive suggestion. We have revised the sentence in Line 33 for clarity. The updated sentence now reads: "Results reveal a clear size-dependence of aerosol hygroscopicity, with the mean hygroscopicity parameter ($\kappa_{mean}$) increased from 0.20 for 30 nm particles to 0.30 for 200 nm particles."

4. Line 57: Do you mean dry deposition efficiency here? Please be explicit.

Response:

Thank you for your comment. In our manuscript, the term "deposition efficiency" specifically refers to the proportion of particles that deposit within the human respiratory tract. This concept focuses on aerosol behavior in the human respiratory system and distinct from atmospheric dry deposition processes.

5. Line 120: Is this division based on the Shi et al 2022? Short explanation on why and how could be also included here in addition to the reference.

Response:

In response to the reviewer's suggestion, we have revised the paragraph as follows: "Given the complex mixing states of ambient aerosols, aerosols are commonly classified into distinct hygroscopic groups based on the hygroscopicity parameter $\kappa$ (Liu et al., 2011), as more-hygroscopic aerosols normally have larger $\kappa$ values and less-hygroscopic particles have smaller ones. In the present work, two different hygroscopic modes were clearly identified. Accordingly, aerosol particles were categorized into two modes with respect to their hygroscopicity: a less-hygroscopic mode (LH, $\kappa \le 0.2$) and a more-hygroscopic mode (MH, $\kappa > 0.2$), consistent with the approach of Shi et al. (2022). The values of $\kappa$ for each mode were calculated as the volume-equivalent mean derived from the $\kappa$-PDF within the respective $\kappa$ boundaries.".

6. Figure 2 and Figure 3: Please make sure the color scale for the k-PDF is perceptually uniform as the current rainbow scale is not and should not be used.

Response:

We thank the reviewer for this important comment. As suggested, we have now replaced the rainbow color scale with a perceptually uniform color scheme (e.g., parula) for all $\kappa$-PDF figures throughout the manuscript. The updated figures are shown below.

[Figure]

**Figure 2. (a-e) Time series of *κ*-PDF for different particle sizes (with the black line indicating the mean *κ* value) and (f-g) Time series of chemical composition of PM₂.₅ obtained using offline and continuous online measurements, respectively, during the sampling period.**

[Figure]

**Figure 3. (a) Average *κ*-PDF for particles at different sizes; (b-f) diurnal variations of the *κ*-PDF for particles of different sizes measured during the campaign.**

7. Sect. 3.2 title: Please avoid using abbreviations in the title, you could use "probability density functions of k" instead. Also please define what it means.

Response:

Thank you for this constructive suggestion. A clear definition of "*κ*-PDF" has been provided upon its first appearance in the revised manuscript, as follows: "The *κ* probability density function (*κ*-PDF), indicative of a statistical distribution that describes the variation in hygroscopicity among an aerosol population, was derived from the GF probability density function (GF-PDF), which was retrieved from the measured GF distribution function (GF-MDF) using the TDMAinv algorithm (Gysel et al., 2009)." In addition, the abbreviation "*κ*-PDF" has been removed from the title of Sect. 3.2.

8. Line 235: Is the abbreviation URG defined somewhere? Even if it is, I would use the full word here too for clarity.

Response:

Thank you for your constructive suggestion. URG refers to URG Corporation. According to the reviewer's suggestion, we have replaced "URG" with "online measurement" in both the figure caption and in the revised manuscript. This change ensures a clearer description of the data source.

9. Figure 4: Are the colored lines/curves in this figure the cluster centroids? This should be stated in the caption. I believe adding a trajectory frequency map for each of the clusters would be very helpful. Please consider adding one to the supplementary material.

Response:

Thanks for your specific comment. In the revised manuscript, we have revised the manuscript accordingly. As suggested, the caption of Figure 5 now explicitly states the colored curves represent the

cluster centroids. In addition, a new figure showing the trajectory frequency for each cluster has been added to the supplementary material.

[Figure]

Figure 4. Cluster analysis of 72 h backward trajectories at 2060 m above ground level at the sampling site during the five trajectory-identified clusters. The line colors denote different clusters, i.e., purple for Cluster 1, yellow for Cluster 2, green for Cluster 3. black for Cluster 4, and blue for Cluster 5.

[Figure]

Figure 5. Frequency distribution of backward trajectories. (Fig.S8 in the SI)

10. Line 245: "was not obviously observed" – what do you mean by this? If you did not measure it, you should say "was not investigated" instead or you could also just say "was not observed" if that's is what you mean.

Response:

As suggested by the reviewer, we modified the sentence into: "However, such an influence on aerosol chemical composition and their hygroscopicity was not observed at the current study".

11. Sect. 3.3/trajectories: You mention trajectory clusters but provide no details on how the clusters were obtained. I am assuming a method of k-mean clustering or similar, however, this should be mentioned in the methods where you first describe your trajectory calculations. The trajectory frequency figure that I suggested to include could then be referenced in the method section already, and later noted in this 3.3 if necessary.

Response:

We thank for the reviewer for this constructive suggestion. In the revised manuscript, we have added a description of the clustering approach in Section 2.4, where trajectory calculations are introduced. In addition, as suggested, we have included a figure showing the trajectory frequency in the supplementary material (see Fig.S8 in the SI) and referenced appropriately. The added text reads as follows: "This study employed the cluster analysis method proposed by Draxler et al (Stein et al., 2015), where the clustering criterion was defined such that the spatial variance of each cluster corresponds to the sum of squared distances between individual trajectories and the mean trajectory of that cluster. The total spatial variance (TSV) was calculated as the sum of the spatial variances of all clusters. The final clustering result was obtained by minimizing the increase in TSV. The trajectory frequency distributions for the resulting clusters are provided in Fig. S8.".

12. Figure 7: I would remove the lined between the points in this figure. You are comparing the hygroscopicity values for different sites and sizes, not necessary looking on how the hygroscopicity changes with size as you can also have particles with different origins (i.e., larger particle is not the smaller one that has grown larger). Please considering increasing both marker and text size in this figure too.

Response:

Thank you for your constructive suggestion. We have removed the lines between the points in Figure 8 (formerly Figure 7) and only showed the individual data points to better highlight the comparisons. Additionally, we have increased both the marker and text sizes in the figure to enhance readability. The updated figure is shown below.

[Figure]

**Figure 6. Comparison of aerosol hygroscopicity measured at different high-altitude sites around the world. (Figure 8 in manuscript)**

13. Supplementary figures & Figure 3a, Figure 5, Figure 7: Please avoid using red and green in the same figure to accommodate color blind readers.

Response:

Thanks for your specific comment. In the revised manuscript, we have adjusted the color scheme of Supplementary Figures, Figure 3a, Figure 5, and Figure 7 to avoid using red and green in the same figure, thereby improving accessibility for color-blind readers.

[Figure]

**Figure 7. (a) Average κ-PDF for particles at different sizes; (b-f) diurnal variations of the κ-PDF for particles of different sizes measured during the campaign. (Figure 3 in manuscript)**

[Figure]

**Figure 8. Cluster analysis corresponding to (a) mean aerosol hygroscopicity parameters κ, (b) the number fraction of LH and MH mode particles, (c) and (d) the mean κ values of LH and MH mode particles at different sizes. (Figure 5 in manuscript)**

[Figure]

**Figure 9. Comparison of aerosol hygroscopicity measured at different high-altitude sites around the world. (Figure 8 in manuscript)**